# CAMELS-NZ: Hydrometeorological time series and landscape attributes for New Zealand

Sameen Bushra<sup>1</sup>, Jeniya Shakya<sup>1</sup>, Céline Cattoën<sup>2,3</sup>, Svenja Fischer<sup>4</sup>, and Markus Pahlow<sup>1</sup>

**Correspondence:** Sameen Bushra (sameen.bushra@pg.canterbury.ac.nz) and Céline Cattoën (Celine.Cattoen-Gilbert@niwa.co.nz)

**Abstract.** We present the first large-sample catchment hydrology dataset for Aotearoa New Zealand with hourly time series: the Catchment Attributes and Meteorology for Large-Sample Studies - New Zealand (CAMELS-NZ). This dataset provides hourly hydrometeorological time series and comprehensive landscape attributes for 369 catchments across New Zealand, ranging from 1972 to 2024. Hourly records include streamflow, precipitation, temperature, relative humidity and potential evapotranspiration, with more than 65% of streamflow records exceeding 40 years in length. CAMELS-NZ offers a rich set of static catchment attributes that quantify physical characteristics such as land cover, soil properties, geology, topography, and human impacts, including information on abstractions, dams, groundwater or snowmelt influences, as well as on ephemeral rivers. New Zealand's remarkable gradients in climate, topography, and geology give rise to diverse hydroclimatic landscapes and hydrological behaviours, making CAMELS-NZ a unique contribution to large-scale hydrological studies. Furthermore, New Zealand's hydrology is defined by highly permeable volcanic catchments, sediment-rich alpine rivers with glacial contributions, and steep, rainfall-driven fast-rising rivers, providing opportunities to study diverse hydrological processes and rapid hydrological responses. CAMELS-NZ adheres to the standards established by most previously published CAMELS datasets, enabling international comparison studies. The dataset fills a critical gap in global hydrology by representing a Pacific Island environment with complex hydrological processes. This dataset supports a wide range of hydrological research applications, including model development and climate impact assessments, predictions in ungauged basins and large-sample comparative studies. The open-access nature of CAMELS-NZ ensures broad usability across multiple research domains, providing a foundation for national water resource and flood management, as well as international hydrological research. By integrating long-term high-resolution data with diverse catchment attributes, we hope that CAMELS-NZ will enable innovative research into New Zealand's hydrological systems while contributing to the global initiative to create freely available large-sample datasets for the hydrological community. The CAMELS-NZ dataset can be accessed at https://doi.org/10.26021/canterburynz.28827644 (Bushra et al., 2025).

<sup>&</sup>lt;sup>1</sup>Department of Civil and Environmental Engineering, University of Canterbury, Christchurch, New Zealand

<sup>&</sup>lt;sup>2</sup>Earth Sciences New Zealand, Christchurch, New Zealand

<sup>&</sup>lt;sup>3</sup>Te Pūnaha Matatini, University of Auckland, Auckland, New Zealand

<sup>&</sup>lt;sup>4</sup>Hydrology and Environmental Hydraulics, Wageningen University & Research, Wageningen, The Netherlands

#### 1 Introduction

55

The study of hydrology has evolved from localised catchment studies to broader approaches encompassing diverse geographical regions and environmental conditions. This shift highlights the importance of large sample hydrology (Falkenmark and Chapman, 1989; Gupta et al., 2014; Andréassian et al., 2006), which analyses data from catchments across various climatic, geological, and land use contexts (McMahon and Peel, 2019). Large-scale datasets are essential for testing hydrological theories and developing models for prediction in ungauged catchments (Gupta et al., 2014; Kuentz et al., 2017; Kratzert et al., 2019a).

One challenge is to ensure the availability of high-quality datasets for the global research community. The Catchment Attributes and Meteorology for Large-Sample Studies (CAMELS) dataset series addresses this by providing comprehensive hydrological data across diverse regions (e.g. Addor et al., 2017; Alvarez-Garreton et al., 2018; Coxon et al., 2020; Fowler et al., 2024; Delaigue et al., 2024). These datasets help to understand catchment behaviour, to refine hydrological models (e.g. Coxon et al., 2019; Kollat et al., 2012; Newman et al., 2015; Perrin et al., 2003) including studies in Aotearoa New Zealand (McMillan et al., 2016), and to examine the impacts of climatic and environmental changes on water resources (Fowler et al., 2021, 2016; Gupta et al., 2014). Large-sample datasets are also crucial for machine learning applications in hydrology, as they provide the necessary high-quality datasets to train models effectively (Slater et al., 2024). The length and quality of calibration data can critically impact the model performance (Ayzel and Heistermann, 2021). Common datasets include streamflow records, meteorological data (e.g. precipitation, temperature, potential evapotranspiration), and static catchment attributes (e.g. topography, soil properties, land cover, geology). This combination allows for in-depth analysis of the relationships of hydrological processes and landscape characteristics, improving predictions (Gauch et al., 2021; Li and Zhao, 2024) and enhancing knowledge on water availability and variability to improve water resources management (Gupta et al., 2014; Addor et al., 2017).

In large-scale hydrology, there is still limited access to data, especially in regions with sparse and inconsistent measurement networks. Crochemore et al. (2020) noted a decline in data availability since the 1980s, with scarce open information in southern Asia, the Middle East, and North and Central Africa. Although the availability of open-access environmental data is increasing, data sources are scattered across various platforms and formats, requiring additional efforts in data collection and pre-processing (Liu et al., 2024). Many state authorities and governmental research institutes now adhere to open data policies, reducing barriers to data sharing (Kibler et al., 2014). But despite efforts to share large-sample datasets globally, hydrometeorological records from different countries often lack consistency, as they are provided in varying formats by different sources. Additionally, these records are rarely compiled at the catchment level, making them less useful for basin-specific research and applications.

The concept of large-sample catchment data development began in the United States with the Model Parameter Estimation Experiment (MOPEX), which provided daily hydrometeorological data from 1948 to 2003 and river characteristics for catchments worldwide (Addor et al., 2017). This dataset was later updated with the latest hydrometeorological time series data (Newman et al., 2015) and extended with additional information on topography, climate, streamflow, land cover, soil, and geol-

ogy, making it well-suited for large-sample studies and comparative hydrology (Addor et al., 2017). Other countries followed the same approach, creating their own large-sample datasets.

The development of new CAMELS datasets (and other diverging names) has covered many geographic regions, including South America (Chile (Alvarez-Garreton et al., 2018); Brazil (Chagas et al., 2020; Almagro et al., 2021)), North America (US (Newman et al., 2015; Addor et al., 2017); Canada (Arsenault et al., 2016); Haiti (Bathelemy et al., 2024)), Europe (UK (Coxon et al., 2020); Denmark (Liu et al., 2024); Switzerland (Höge et al., 2023); Saxony in Germany (Hauffe et al., 2023); Germany (Loritz et al., 2024); France (Delaigue et al., 2024); Iceland (Helgason and Nijssen, 2024); Sweden (Teutschbein, 2024)), Africa (Tramblay et al., 2021), and Asia (China (Hao et al., 2021); India (Mangukiya et al., 2025)). While datasets for Australia have been compiled (Fowler et al., 2024, 2021), there remains a gap in the Oceania region.

The objectives of developing these datasets vary. For example, Great Britain aimed to integrate diverse information into a single, consistent, and up-to-date dataset (Coxon et al., 2020). CAMELS-AUS focuses on studying Australia's unique hydrology, particularly arid regions, low rainfall-to-runoff ratios, and long-term droughts (Fowler et al., 2021). CAMELS-CH (Switzerland) addresses climate change impacts like glacier retreat, snow cover reduction, and altered streamflow patterns, helping researchers understand and predict shifts in the hydrological cycle (Höge et al., 2023). CAMELS-US aims to develop a large hydrometeorological dataset for analysing regional differences in hydrological model performance across the contiguous United States, serving as a community resource for improving predictions and exploring key scientific questions about basin behaviour under diverse hydroclimatic conditions (Addor et al., 2017).

65

80

CAMELS-NZ (Bushra et al., 2025) is designed to address the unique hydrological challenges of its region, providing a crucial resource for understanding and managing New Zealand's diverse and dynamic water systems. The CAMELS-NZ dataset provides hydrometeorological time series and landscape attributes for 369 catchments across New Zealand, aligning with similar datasets from other countries to support large-sample hydrology. A unique feature of CAMELS-NZ is its high temporal resolution, offering hourly records of streamflow, rainfall, temperature, relative humidity and potential evapotranspiration, along with detailed catchment attributes such as land cover, soil particle size, and geology. The inclusion of hourly streamflow data is particularly valuable for capturing the fast dynamics of hydrological processes, especially in steep catchments, and supports the development of high-resolution models for applications such as flood risk management. By facilitating comparative hydrology at both national and global scales, CAMELS-NZ addresses a critical gap in hydrological modelling to predict and manage water systems amid climate variability and environmental change (Gupta et al., 2014; Kratzert et al., 2019b). This is crucial for New Zealand, which faces challenges in water management, including balancing supply and demand, mitigating flood risks (Cattoën et al., 2025; Harrington et al., 2023; Johnson et al., 2025), and adapting to a changing climate (Mullan et al., 2012; Collins, 2020b).

## 2 Objectives

90

New Zealand's remarkable gradients in climate, topography, and geology (Collins, 2020a; McMillan et al., 2016; Snelder and Biggs, 2002) give rise to a diverse array of hydro-climatic landscapes, each offering unique opportunities for hydrological studies. For CAMELS-NZ, three key characteristics are particularly noteworthy.

Firstly, New Zealand's inclusion in the CAMELS initiative brings a distinct hydrological perspective to the global pool of CAMELS dataset. The CAMELS-NZ dataset encompasses up to 52 years of hydrometeorological data (1972-2024), capturing a wide range of extreme events, including Cyclones Bola (1988), Debbie and Cook (2017), Auckland Anniversary floods (2023), and Cyclone Gabrielle (2023), alongside significant atmospheric river events (Kingston et al., 2016) such as those in Canterbury (2021), Westport (2021) and Nelson (2022). Coupled with high-resolution hourly flow data, CAMELS-NZ is uniquely positioned to study the dynamics of fast-rising rivers, driven by New Zealand's steep topography and intense rainfall events. These systems, often associated with flash floods, are currently under-represented in existing CAMELS datasets, yet play a critical role in understanding rapid hydrological responses and improving global flood forecasting models. In addition, major drought events like the one in 2010, where only a third of the mean annual rainfall was observed, or the 2008 drought in the Waikato area are included in the data set. This makes the dataset a useful resource for investigating fast hydrological responses and hydrological regimes under both historical and evolving climatic conditions, while addressing broader issues related to the water cycle and water resource management.

Secondly, New Zealand's catchments offer valuable insights into hydrological processes typical for Pacific Island nations, e.g. volcanic catchments. These catchments are characterised by unique geomorphological features, such as a volcanic plateau with very high rates of infiltration, braided rivers, sediment loads, and glacial melt contributions from alpine regions. New Zealand's hydrology is shaped by the interplay of tectonic activity, steep gradients, and climate variability. Incorporating these catchments into the CAMELS data pool offers an opportunity to study hydrological processes in volcanic and alpine environments, complementing datasets from regions such as Chile (Alvarez-Garreton et al., 2018) and Switzerland (Höge et al., 2023), while filling a critical gap in the Pacific region.

Third, the CAMELS-NZ dataset is designed to provide a large-sample catchment dataset for New Zealand, overcoming the limitations of existing fragmented datasets in the country. By combining hydrometric data, multiple sources of catchment characteristics (Snelder and Biggs, 2002; Leathwick et al., 2010), and qualitative metadata (Cattoën et al., 2025), it offers researchers a comprehensive and accessible resource to study New Zealand's catchments on a national scale. It goes beyond global data sets such as ROBIN (Turner et al., 2025), which also include catchments in New Zealand (in ROBIN 111 catchments are included), but either have a lower temporal resolution and/or include fewer catchments than the current CAMELS-NZ.

#### 115 3 New Zealand physical and hydroclimatic context

New Zealand is a country comprised of two main islands, Te Ika-a-Māui the North Island and Te Waipounamu the South Island, each with distinct and unique geographical features. The South Island topography is dominated by the Southern Alps, which stretch over 500 kilometres and rise to altitudes exceeding 3,700 metres (Spronken-Smith and Sturman, 2001). In contrast, the

North Island features a volcanic plateau and geothermal areas. The mean elevation of the North Island is 298 m, while that of the South Island is 627 m (Figure 1). These distinct features of the two islands contribute to New Zealand's extraordinary hydrometeorological conditions, impacting everything from weather patterns and rainfall distribution to climate variations and sunshine hours (Cattoën et al., 2025).

The hydrometeorological conditions of the North and South islands vary significantly. In the South Island, annual precipitation can reach more than 15,000 mm (Henderson and Thompson, 1999; Kerr et al., 2011; Cattoën et al., 2025) in the West Coast region, while the driest areas receive as little as 350 mm (Henderson and Thompson, 1999; Kerr et al., 2011; Kingston et al., 2016). The North Island experiences a range of mean annual rainfall from 500 mm in drier regions to 3,000 mm in wetter areas (Caloiero, 2014). Temperatures also vary, with mean annual values ranging from 10°C in the south to 16°C in the north of New Zealand. Most of the country receives at least 2,000 hours of sunshine per year, and during summer, the midday UV index (UVI) frequently reaches very high or extreme levels, particularly in northern and mountainous regions. Snowfall is predominantly confined to mountain regions, with coastal areas of the North Island and the western South Island rarely experiencing snow. In contrast, the eastern and southern parts of the South Island occasionally receive snowfall during winter (Kerr et al., 2011). These climatic variations give rise to diverse hydrological conditions across New Zealand.

New Zealand's landscapes have undergone significant changes over time, shaped by natural processes and human activities such as deforestation, grassland conversion, and urbanisation. These factors, combined with the country's geography and location, make it particularly vulnerable to natural disasters, including earthquakes, volcanic eruptions, erosion, landslides, and extreme weather events (EHINZ, 2024). Among these, earthquakes and tsunamis pose some of the greatest risks due to New Zealand's position along the tectonic boundary between the Australian and Pacific Plates. However, weather-driven hazards such as storms and floods occur most frequently, with climate change intensifying their impacts (Hendrikx and Hreinsson, 2012). These extreme events also contribute to widespread soil erosion which is a persistent issue in New Zealand. The country experiences one of the highest erosion rates globally, transporting approximately 200 million tonnes of sediment to the ocean per year (Hicks et al., 2011).

Climate change is already reshaping New Zealand's environment, with clear indicators such as variability in snow and glacier mass balance (Harrington et al., 2023; Hopkins et al., 2015; Hendrikx and Hreinsson, 2012). New Zealand's climate has warmed by +0.91°C from 1909 to 2009, as shown by the nationally representative Seven Station Series (Mullan et al., 2010). Recent analyses also indicate significant shifts in temperature and precipitation normals across regions and seasons, reflecting both natural variability and anthropogenic climate change (Srinivasan et al., 2024). Recent high-resolution (~12 km) downscaled climate projections for New Zealand, based on six global climate models (GCMs) and three regional climate models (RCMs), project a national annual mean warming of 3.1°C (range: 2.0–3.8°C) by 2080–2099 relative to 1986-2005, under the high-emissions SSP3-7.0 scenario (Gibson et al., 2025). Summer maximum temperatures are expected to increase by 3.9°C on average (range: 2.8–4.8°C), particularly affecting inland North Island and high-elevation areas (Gibson et al., 2025). These findings are qualitatively consistent with earlier CMIP5-based projections (Mullan et al., 2018; Gibson et al., 2025). Precipitation projections show a distinct seasonal and spatial pattern, with winter and spring rainfall increasing by over 20% in parts of the South Island's west coast, while northern and eastern regions of the North Island are likely to experience reduced

Figure 1. Elevation across the different regions of New Zealand.

rainfall, especially in spring and summer (Gibson et al., 2025). Across much of the country, extreme precipitation events are expected to become more intense but less frequent, occurring over shorter durations – except on the South Island's west coast,

where both totals and extremes increase, likely due to topographically enhanced dynamical processes (Gibson et al., 2024). Overall, these projected changes reinforce trends identified in earlier CMPI5-based assessments (Gibson et al., 2025).

Weather-triggered hazards are significantly influenced by climate change, as all hydrological parameters are subject to change, such as rainfall patterns, temperature, and river flows (Gibson et al., 2025). In New Zealand, headwaters of most major rivers are fed from mountains (Henderson and Thompson, 1999). These rivers exhibit highly variable flow regimes and are prone to large floods relative to their catchment size (Jowett and Duncan, 1990). The concentration times for flood peaks are often less than 12 hours, highlighting the rapid hydrological response to precipitation (Griffiths and McSaveney, 1983).

## 4 Data governance and providers

New Zealand's hydrometric data collection began in the 1900s as a coordinated nationwide network. However, subsequent legislative changes (Pearson, 1998) have led to a fragmented network. Today, each Regional Council funds and maintains its own hydrometric network and models based on local priorities, while at the national scale, the Earth Sciences New Zealand (formerly known as the National Institute of Water and Atmospheric Research (NIWA)) maintains a limited national hydrometric network (Pearson, 1998; Cattoën et al., 2025).

Despite these efforts, a nationwide, consistent and freely available dataset, such as the one presented here, has not previously been available for New Zealand. During the International Hydrological Decade (1965-74), 50 out of 90 representative river catchments were monitored, which was considered sufficient to capture New Zealand's diverse hydrological conditions (Toebes and Palmer, 1969). However, hydrometeorological data today remain dispersed across multiple institutions using different systems. Earth Sciences New Zealand provides climate, river flow, and environmental monitoring data; MetService, New Zealand's weather service, handles weather forecasting and climate data; Regional Councils manage water quality and flood monitoring at a local level; furthermore, Earth Sciences New Zealand also focuses on groundwater, geological, and seismic data.

To address these challenges, the CAMELS-NZ dataset was developed to integrate New Zealand's hydrometeorological data into a standardised and comprehensive resource. By reducing data fragmentation and inconsistency, this dataset can help to enhance model calibration and uncertainty assessment, improve national and regional predictions, and support more effective water resource management. Its high temporal resolution allows for the consideration of dynamic processes and fast response times in catchments, which is particularly important given New Zealand's hydrological variability. The goal of CAMELS-NZ is to provide a robust basis to help address current and future challenges related to climate change and its associated risks for people, industry, primary production, and the environment.

CAMELS-NZ compiles data from different sources, including Regional Councils, companies, private operators, the National Institute of Water and Atmospheric Research (McMillan et al., 2016; Cattoën et al., 2025), River Environment Classification (REC) (Snelder and Biggs, 2002), and Freshwater Ecosystems of New Zealand (FENZ) (Leathwick et al., 2010). The REC framework categorises river environments based on key physical and environmental characteristics such as climate, topography, geology, and land cover. FENZ is a geospatial database that provides detailed information about New Zealand's freshwater

environments, supporting conservation, management, and research. By incorporating these diverse datasets, CAMELS-NZ creates a unified, high-quality resource, enabling an improved understanding and management of New Zealand's water resources.

#### 5 Catchments characteristics

A total of 369 catchments have been selected, distributed relatively uniformly across New Zealand. These catchments were selected based on data availability and quality criteria, as described in the following section. In addition, the availability of catchment attributes was considered, with time series from stations lacking catchment data being excluded. The record for these catchments begins in 1972 as continuous climate data were only available from this time onward.

New Zealand exhibits significant variability in catchment characteristics. The catchment areas range from small upstream tributaries of a few square kilometres to large river systems (Figure 2a), with areas classified into bands of up to 6,000 km<sup>2</sup>. Larger upstream areas are more prevalent in the South Island. The steepness of catchments varies considerably from flat lowlands to steep alpine regions, especially in the South Island (Figure 2b). While most catchments have moderate slopes, some are either extremely steep or entirely flat. The South Island is home to a greater concentration of high-elevation catchments due to the Southern Alps, while the North Island primarily consists of lowland areas with some elevated regions such as the central volcanic plateau. Stream order categorises the hierarchy of stream networks (Figure 2d). Lower-order streams are widely distributed across both islands, reflecting the abundance of small tributaries. Higher-order streams, on the other hand, are concentrated in regions with large river systems, particularly in the South Island. These attributes collectively illustrate New Zealand's varied hydrological landscape (Table 1).

Table 1: Catchment attributes.

| Attribute name    | Description                               | Unit            | Source                                                                 |
|-------------------|-------------------------------------------|-----------------|------------------------------------------------------------------------|
| Station_ID        | Station Identifier                        | -               |                                                                        |
| RID               | Reach ID (alternative ID)                 | -               |                                                                        |
| Station Name      | Name of the station where gauge is placed | -               | <ul><li>Station Information</li><li>Management System (SIMS)</li></ul> |
| Latitude (WGS 84) | Latitude of gauge                         | degree          |                                                                        |
| Longitude(WGS 84) | Longitude of gauge                        | degree          |                                                                        |
| uparea            | Catchment area upstream of outlet         | $\mathrm{km}^2$ |                                                                        |
| Region            | Administrative regions of NZ              | -               | <ul><li>Freshwater Ecosystem</li></ul>                                 |
| UpStreamLakes     | The presence of a lake upstream           | -               | of New Zealand (FENZ)                                                  |
|                   |                                           |                 | —<br>Geodatabase                                                       |

| usLake       | Lake Index quantifies the influence of upstream lakes on river flow and hydrology (proportion of a river's upstream catchment area that is covered by lakes, affecting flow regulation, sediment transport, and nutrient retention, higher Lake Index indicates greater lake influence, typically leading to more stable flow regimes) | ratio               |                                                      |
|--------------|----------------------------------------------------------------------------------------------------------------------------------------------------------------------------------------------------------------------------------------------------------------------------------------------------------------------------------------|---------------------|------------------------------------------------------|
| Stream_Order | Strahler order at the outlet                                                                                                                                                                                                                                                                                                           | -                   | -                                                    |
| elevation    | Catchment average elevation                                                                                                                                                                                                                                                                                                            | m a.s.1.            |                                                      |
| usSteep      | Proportion of catchment with slope $< 30^{\circ}$ (not steep)                                                                                                                                                                                                                                                                          | %/100               | Multiscale River Environment Classification for      |
| usLowGrad    | Proportion of catchment with slope $>30^\circ$ (steep), indicator of erosion and flow variability                                                                                                                                                                                                                                      | %/100               | Water Resources Management (Snelder and Biggs, 2002) |
| usAveSlope   | Average slope of catchment calculated from 30 m DEM grid                                                                                                                                                                                                                                                                               | angle - de-<br>gree |                                                      |
| DIST_SEA     | Distance to the sea from the gauge station                                                                                                                                                                                                                                                                                             | m                   | -                                                    |
| SRC_OF_FLW   | Source of flow describes hydrological and sediment transport processes within a catchment (Mountain: > 50% annual rainfall volume above 1000 m; Hill: 50% annual rainfall volume between 400 and 1000 m; Low Elevation: 50% rainfall volume below 400 m; Lake: Lake influence index > 0.033)                                           | -                   | _                                                    |
| Records      | Total number of years of time series data                                                                                                                                                                                                                                                                                              | years               | Calculated from time series data                     |

| Landcover | Categorical variable describing the - | Freshwater Ecosystem  |
|-----------|---------------------------------------|-----------------------|
|           | spatially dominant landcover: Bare    | of New Zealand (FENZ) |
|           | ground (B), Exotic forest (EF), In-   | Geodatabase           |
|           | digenous forest (IF), Pastoral (P),   |                       |
|           | Scrub (S), Tussock (T) and Urban      |                       |
|           | (U)                                   |                       |

# 6 Hydrometerological Time Series

Hourly hydrometeorological time series data are provided for the 369 CAMELS-NZ catchments including streamflow, precipitation, temperature, relative humidity, and potential evapotranspiration. These datasets were chosen for inclusion in CAMELS-NZ to cover the common forcing and evaluation data needed for catchment hydrological modelling and analysis (Figure 3). All time series data are reported in New Zealand Standard Time (NZST) which is UTC +12 hours. An overview is given in Table 2 with related climatic attributes given in Table 3.

Table 2: Hydro-meteorological time series.

| Attribute name    | Description                         | Unit       | Source                                         |
|-------------------|-------------------------------------|------------|------------------------------------------------|
| Flow              | Hourly flow                         | $m^3/s$    | Earth Sciences NZ, Regional                    |
|                   |                                     |            | Councils, District Council, City               |
|                   |                                     |            | Councils                                       |
| Precipitation     | Hourly rainfall                     | mm         | Tait et al. (2006b), Tait et al.               |
|                   |                                     |            | (2012), Rupp et al. (2009b)                    |
| PET               | Hourly potential evapotranspiration | mm         | McMillan et al. (2016),<br>Clark et al. (2008) |
| Temperature       | Hourly temperature                  | Kelvin     |                                                |
| Relative_humidity | Hourly relative humidity            | Percentage |                                                |

Table 3: Climatic attributes.

| Attribute name | Description                                         | Unit      | Source                                     |
|----------------|-----------------------------------------------------|-----------|--------------------------------------------|
| usDaysRainGT25 | Total annual catchment rain days greater than 25 mm | Days/Year | Freshwater Ecosystem of New Zealand (FENZ) |
|                |                                                     |           | Geodatabase                                |

| usRainDays10      | Average number of days per year      | mean       |                              |
|-------------------|--------------------------------------|------------|------------------------------|
|                   | within catchment with daily rain-    | #days/year |                              |
|                   | fall greater than 10 mm, calculated  |            |                              |
|                   | from the monthly counts and aver-    |            |                              |
|                   | aged over the year                   |            |                              |
| usAnRainVar       | Coefficient of variation of annual   | Percentage |                              |
|                   | catchment rainfall                   |            |                              |
| Mean Annual Rain- | Average total precipitation per year | mm         |                              |
| fall              | over the observation period          |            |                              |
| usPET             | Annual potential evapotranspiration  | mm         |                              |
|                   | of catchment                         |            |                              |
| Climate Zone      | Climate zones defined for New        | -          | Multiscale River Environment |
|                   | Zealand: cool dry, cool wet, cool    |            | Classification for Water Re- |
|                   | extremely wet, warm dry, warm wet    |            | sources Management (Snelder  |
|                   | and warm extremely wet               |            | and Biggs, 2002)             |

#### 6.1 Streamflow

Streamflow data is sourced from the Earth Sciences NZ, regional councils, companies and private operators, ensuring comprehensive coverage across both the North and South Islands. Hourly streamflow records are available with varying lengths between the years 1972 to 2024 (Figure 3). Figure 3 illustrates the spatial and temporal patterns of data availability across hydrological stations. The main map visualises the stations, colour-coded based on their record duration. This figure provides insights into the density and temporal coverage of monitoring locations across the country. Overall, 96% (355) of gauges have at least 20 years of data, and 67% (248) have at least 40 years of data. The inset graph in the upper-left corner illustrates the percentage of available data over time for different decadal monitoring periods (e.g., 1972–2024, 1980–2024, etc.). The curves show a general decline in the number of stations as the percentage of complete data increases, indicating that fewer stations maintain long-term, uninterrupted records. The analysis of available data indicates data gaps over time periods, with older records (e.g., 1972–2024) having more significant gaps compared to more recent datasets (e.g., 2010–2024). Hourly data for precipitation, streamflow, temperature, relative humidity, and potential evapotranspiration were aggregated to daily values. Daily totals were calculated for precipitation, while daily means were computed for streamflow, temperature, relative humidity, and PET.

Data quality was checked for all discharge time series. Calculated flow statistics, such as Mean Annual Flood (MAF) and Mean Hourly Flow (MHF), were cross-verified with reference statistics provided by a previous Earth Sciences NZ Flood statistics study (Roddy Henderson, 2016). The MAF and MHF were plotted against catchment area to identify outliers. These plots

Figure 2. (a) Upstream catchment area (b) slope (c) Elevation (d) Stream order.

are effective for assessing data quality since streamflow generally scales with upstream area under natural conditions. Significant deviations from this expected relationship may indicate data errors. All streamflow time series were visually inspected for errors, gaps, and the plausibility of the hydrographs (Alfieri et al., 2013; Coxon et al., 2020; Crochemore et al., 2020; Delaigue et al., 2024). Time series affected by, e.g., changes of the measurement system or in the rating curve have been excluded.

# 6.2 Meteorological time series

New Zealand shows high spatial variability in precipitation, in particular in its mountainous regions. Although a very dense gauging network would be needed to capture the high spatial variability in rainfall for the mountainous regions of New Zealand, such a network currently does not exist. Moreover, due to the complex terrain in a range of regions, radar accuracy can be compromised, and coverage is often limited (Cattoën et al., 2020). However, daily precipitation data are accessible across New Zealand through the interpolated and mass-corrected Virtual Climate Station Network (VCSN) (Tait et al., 2006b). This

**Figure 3.** Length of streamflow records for each catchment; the line plot shows the proportion of catchments with data gaps, differentiated by the observation period.

daily dataset developed by Earth Sciences NZ includes other key variables such as temperature and relative humidity (Tait and Woods, 2007). While the VCSN offers valuable national coverage, previous evaluations have noted interpolation errors in areas of complex terrain due to sparse observational coverage (Tait and Macara, 2014; Tait et al., 2012). Additionally, the number of contributing stations has declined significantly overtime, ranging from a peak of approximately 1,400 stations between 1970 and 1986 to fewer than 400 stations in recent years. This decline is primarily due to the discontinuation of data collection by individual farmers as well as the gradual takeover of stations by local authorities, which has led to reduced station density in several regions.

VCSN provides interpolated and mass-corrected daily precipitation values at a 5-km spatial resolution across the country (Tait et al., 2006a). The dataset interpolates observed meteorological values from the rain gauge network ((McMillan et al., 2016; Tait et al., 2006b, 2012)). To enhance spatial accuracy, it accounts for elevation and climate gradients using a thin plate smoothing spline interpolation method, which incorporates latitude, longitude, and a third pattern variable, interpolated mean

annual rainfall (Tait et al., 2006a). To provide hourly precipitation data that match the hourly discharge series, the time series of precipitation were derived using an inverse square interpolation of the VCSN rainfall totals to the centroid of each subcatchment and temporally disaggregated (McMillan et al., 2016). The daily-to-hourly disaggregation process involves using multiplicative random cascades (MRCs) to calculate the proportion of daily rainfall occurring in each hour from nearby automatic rain gauges Rupp et al. (2009a). These hourly proportions are then spatially interpolated to the centroid of each catchment and multiplied by the total daily rainfall, ensuring consistency with the original daily totals (McMillan et al., 2016; Clark et al., 2008). Finally, the bias-corrected and catchment averaged gridded rainfall data is considered for the dataset.

The same procedure, using catchment-average values, was applied to the daily temperature, whereby the mean was taken from daily maximum and minimum VCSN temperatures. Relative humidity was disaggregated into hourly values by first calculating the dew point temperature at 9 AM based on the daily relative humidity and air temperature at that time. This dew point was then uniformly distributed across all 24 hours unless it exceeded the air temperature at any given hour, in which case it was set equal to the air temperature to maintain physical consistency. Potential evapotranspiration (PET) was calculated using a modified Priestley–Taylor method (Priestley and Taylor, 1972), following the implementation in McMillan et al. (2016) and Clark et al. (2008). Net radiation was estimated as the sum of net shortwave radiation (adjusted for albedo) and net longwave radiation, computed empirically using air temperature, vapour pressure (from dew point temperature), and a cloudiness factor (Shuttleworth, 1993). The ground heat flux term was assumed negligible. The slope of the saturation vapour pressure curve and the psychrometric constant were calculated using Equations 4.2.3 and 4.2.28 in Shuttleworth (1993), both of which are explicit functions of air temperature. The Priestley–Taylor coefficient ( $\alpha$ ) was set to 1.26, appropriate for humid conditions. While relative humidity was not used directly, it affects vapour pressure as dew point is derived from relative humidity. This approach offers a computationally efficient and physically grounded estimate of PET, suited to large-scale hydrological modelling in New Zealand's climatic conditions.

Figure 4 presents the spatial distribution of (a) mean annual rainfall in mm, (b) annual rainfall variability in %, and (c) mean annual potential evapotranspiration (PET) in mm across the CAMELS-NZ catchments. Relative rainfall variability is highest in the eastern regions of both islands, particularly in the South Island, where lower annual rainfall is observed. In contrast, the western regions exhibit lower relative variability but much higher rainfall totals, as indicated in the mean annual rainfall (Figure 4a). This reflects New Zealand's distinct west to east hydroclimatic gradient, where prevailing westerlies and orographic effects produce consistently high rainfall in the west and more variable, drier conditions in the east (Tait et al., 2006b). Annual PET shows a similar pattern, with higher values to the north and east and very low values to the west of the Southern Alps (Figure 4c).

# 7 Catchment attributes

To provide general catchment information, we include catchment attributes describing location, topography, climatic indices, geology, land cover, and anthropogenic attributes. Tables 1, 4 and 5 present an overview of all catchment attributes, including units and data sources.

**Figure 4.** (a) Mean Annual rainfall in mm (b) Coefficient of variation of annual catchment rainfall in % (c) Annual potential evapotranspiration in mm.

Table 4: Geological attributes.

| Attribute name | Description                        | Unit    | Source                |
|----------------|------------------------------------|---------|-----------------------|
| usParticleSize | Catchment average of particle size | Ordinal | Freshwater Ecosystem  |
|                | of underlying rocks, 1 (fine) to 5 |         | of New Zealand (FENZ) |
|                | (coarse)                           |         | Geodatabase           |
| usHard         | Catchment average of hardness of   | Ordinal |                       |
|                | underlying rocks, 1 (low) to 5     |         |                       |
|                | (high)                             |         |                       |

| usCalc  | Catchment average of calcium (Index)                                                                                                                                                                                     | Ordinal | Multiscale River Environment Classification for         |
|---------|--------------------------------------------------------------------------------------------------------------------------------------------------------------------------------------------------------------------------|---------|---------------------------------------------------------|
| Geology | Categorical variable describing the spatially dominant geology: Alluvium (Al), Hard sedimentary rocks (HS), Soft sedimentary rocks (SS), Volcanic basic (VB), Volcanic acidic (VA), Plutonics (PI) and Miscellaneous (M) | -       | Water Resources Management<br>(Snelder and Biggs, 2002) |

Table 5: Anthropogenic and other flow influencing attributes.

| Attribute name   | Description                                                                                                                                                                                                                          | Unit   | Source                |
|------------------|--------------------------------------------------------------------------------------------------------------------------------------------------------------------------------------------------------------------------------------|--------|-----------------------|
| Influence        | Factors influencing the catchment such as abstraction, flow control,                                                                                                                                                                 | -      |                       |
|                  | natural influences or other reasons                                                                                                                                                                                                  |        | _                     |
| IsDam            | Parameters related to                                                                                                                                                                                                                |        | Cattoën et al. (2025) |
| IsWeir           | anthropogenic or other influences                                                                                                                                                                                                    | Yes/No |                       |
| IsAbstracted     | on catchment obtained through questionnaire survey (presence of dam, weir, abstraction, ephemeral rivers, groundwater influence, unreliable periods of ratings exist, influence of snow, gauges installed for specific flow regimes) |        |                       |
| IsEphemeral      |                                                                                                                                                                                                                                      |        |                       |
| IsGWinfluenced   |                                                                                                                                                                                                                                      |        |                       |
| IsRatingOk       |                                                                                                                                                                                                                                      |        |                       |
| IsSnowInfluenced |                                                                                                                                                                                                                                      |        |                       |
| DSDamAffected    |                                                                                                                                                                                                                                      |        | _                     |
| Flow Regime      | Monitoring purpose (All flows,                                                                                                                                                                                                       | -      |                       |
|                  | Flood warning only, Low to mid-                                                                                                                                                                                                      |        |                       |
|                  | range (e.g. water resource consent),                                                                                                                                                                                                 |        |                       |
|                  | Except low flow)                                                                                                                                                                                                                     |        |                       |

New Zealand's climate is highly varied, ranging from warm subtropical zones in the north to cool temperate climates in the south, with severe alpine conditions in the mountainous areas. The country's mountain chains create distinct climate zones by

blocking the prevailing westerly winds, resulting in contrasting conditions on either side. CAMELS-NZ provides the climate zones (Figure 5d), where CD is cool dry, CW is cool wet, CX is cool extremely wet, WD is warm dry, WW is warm wet and WX is warm extremely wet, derived from REC. The West Coast of the South Island falls under the CX (cool extremely wet) classification, receiving the highest rainfall, while areas to the east (CD, cool dry) are much drier. Seasonal variations in rainfall include that northern and central regions experience more winter rainfall, while southern areas have the least rainfall in winter.

Mean catchment temperatures range from 10°C in the south (CW cool wet) to 16°C in the north (WD warm dry and WW warm wet), with inland and eastern areas showing greater seasonal temperature variations.

CAMELS-NZ also provides a classification of the underlying geology for each catchment, using data from REC. New Zealand, a geologically young country (Pettinga, 2001), features diverse geology classified as Alluvium (Al), Hard sedimentary rocks (HS), Soft sedimentary rocks (SS), Volcanic basic (VB), Volcanic acidic (VA), Plutonics (PI), and Miscellaneous (M) (Figure 5a). The oldest rocks are found along the South Island's west coast and mountain ranges, and predominantly consist of HS and PI (hard sedimentary rocks HS and plutonics PI). The Taupo Volcanic Zone (VA), extending from the central plateau to the northeast coast, has porous pumice/tephra deposits over ignimbrite sheets (Neall, 2001), where very high rates of infiltration to the underlying aquifers lead to strongly damped flow responses with high baseflow. Outwash plains (Al) comprise alluvial gravels, forming major aquifers. In these Al regions, layered confined aquifers cause groundwater losses upstream and gains downstream (Yang et al., 2017). For the considered data sets, only information on the dominant geological class in the respective catchment is available. There does exist another geology data base provided by GNS-Science (2012) with gridded data which could be used to obtain more detailed information on the exact percentage of each geological class. However, these classes deviate from the current ones and were thus not included in CAMELS-NZ.

Land cover data includes information on forest cover, agricultural land use, urbanisation, and other significant land cover types. Seven major categories of land cover are used to differentiate New Zealand, which are bare ground (B), exotic forestry (EF), indigenous forest (IF), pastoral (P), scrub (S), Tussock (T) and urban (U) category (Figure 5b). Native vegetation (IF, S, T) covers 47% of New Zealand, including beech forests, mixed forests, scrub, and native grasslands, with some bare ground (B) and glacial areas in the higher mountains of the South Island (McMillan et al., 2016). The remaining land is largely agricultural, with 40% in pastoral farming (P) and 7% in exotic forestry (EF). Urban areas (U) account for just 1% of land cover.

The anthropogenic catchment attributes in the CAMELS-NZ dataset were derived from a previous study Cattoën et al. (2025). The classification is based on data from questionnaires completed by experts and stakeholders, providing essential information on human influences on hydrological processes within the catchments, and the monitoring purpose of a site ('Flow Regime'). The anthropogenic attributes capture various modifications to natural water systems, such as the presence of dams, water abstraction, and other human interventions that impact streamflow and flood behaviour. Each attribute, whether it is the presence of dams, weirs, water abstraction, or the influence of groundwater and snowmelt, serves as an important indicator of how human activities modify the natural hydrology of a catchment. These attributes offer a clear assessment of the types of modifications present, their potential impacts, and their spatial distribution across New Zealand's diverse landscapes (Figure 5 c).

Figure 5. Catchment attributes for New Zealand included in the CAMELS-NZ data set; (a) Geology: Alluvium (Al), Hard sedimentary rocks (HS), Soft sedimentary rocks (SS), Volcanic basic (VB), Volcanic acidic (VA), Plutonics (PI) and Miscellaneous (M); (b) Land cover: Bare ground (B), Exotic forestry (EF), Indigenious forest (IF), Pastoral (P), Scrub (S), Tussock (T) and Urban category (U); (c) Type of influence; (d) Climate zones defined for New Zealand (CD - cool dry, CW - cool wet, CX - cool extremely wet, WD - warm dry, WW - warm wet and WX - warm extremely wet)

# 8 Conclusions

The CAMELS-NZ dataset introduced here provides the first high-resolution, open-source catchment dataset for New Zealand. It provides hourly time series of hydrological and meteorological data, alongside a wide array of catchment attributes, including geology, land use, climate zones, and anthropogenic influences. Designed to capture New Zealand's remarkable gradients in climate, topography, and geology, the dataset supports applications such as climate change impact assessments, water resource planning, or flood and drought analysis and management.

New Zealand's inclusion in the global CAMELS initiative adds a distinct hydrological perspective, leveraging more than 50 years of hydro-meteorological data that capture diverse extreme events. With its high-resolution hourly flow data, the dataset is particularly suited to studying fast-rising rivers driven by steep topography and intense rainfall. These systems, often associated with flash floods, are under-represented in existing CAMELS datasets and provide critical insights for advancing local and global flood forecasting models. Additionally, the availability of long-term records facilitates statistical analyses and modelling efforts.

New Zealand's volcanic and alpine catchments further enrich the dataset, offering insights into hydrological processes typical of Pacific Island nations. These catchments feature unique geomorphological characteristics, such as volcanic plateaus with high infiltration rates, braided rivers, and glacial melt contributions from alpine regions. By capturing these diverse systems, CAMELS-NZ complements other CAMELS datasets, such as those for Chile and Switzerland, while filling a critical gap in the Pacific region.

The CAMELS-NZ dataset consolidates previously fragmented hydrometric data, catchment attributes, and qualitative metadata into a comprehensive, accessible resource. This integrated dataset provides researchers with new opportunities to study New Zealand's catchments at a national scale, fostering advancements in hydrological process understanding, catchment modelling, and water resources planning and management.

The current dataset is based on REC version 1, as it provides the most complete set of catchment attributes. Future updates should incorporate more recent REC versions, and efforts should focus on expanding the dataset and enhancing its applicability for diverse hydrological studies.

#### 9 Data availability

335

The dataset can be accessed at https://doi.org/10.26021/canterburynz.28827644 (Bushra et al., 2025). Some streamflow data require permission from the data provider. This applies to a total of 14 stations. Information on how to obtain permission is provided in the readme file. Streamflow files that require permission are left empty, while all other related data such as precipitation, PET and catchment attributes are provided. Please also note that some data come with limitations. These are listed in the readme file.

Author contributions. SB and JS collated and plotted the data with the guidance of CC. SB, JS and CC quality-controlled the data. SB, JS, CC, SF and MP wrote the draft version of the manuscript. All authors reviewed and edited the manuscript.

Competing interests. No competing interests are present.

Acknowledgements. The authors gratefully acknowledge all the data owners, collectors, processors and providers who made this dataset possible. SB was supported by the University of Canterbury Sustainable Development Goals PhD scholarship. JS was supported by the University of Canterbury Department of Civil and Natural Resources Engineering PhD scholarship. CC was supported by the Ministry of Business and Innovation and Employment under contract C01X1703. The University of Canterbury Library Open Access Fund provided funding to publish open access.

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
