# Peer review of "CAMELS-NZ: Hydrometeorological time series and landscape attributes for New Zealand"

_Earth System Science Data, 2025_

## Author Comment (AC1)

Thank you very much for providing detailed comments, which allowed us to refine the manuscript and the dataset.

Please see the comments provided by the referee in black font, and our point-by-point response in blue font.

**Anonymous Referee #1:**

This paper describes a new open access dataset that will be of great value to hydrologists and others in the earth sciences. It is clear and well written and includes excellent background on NZ's climate, landscape and geology. I recommend that the paper is published if the following minor comments are addressed satisfactorily.

We would like to thank Referee #1 for the positive feedback and the insightful comments. Please find our detailed responses to each point below (blue font).

1. Lines 140-145: I would recommend using a more recent source for future estimates of temperature and rainfall changes instead of King 2010 as this is based on NIWA's modelling done nearly 20 years ago for AR4. For instance:

   Peter Gibson, et al., 2025, Downscaled CMIP6 future climate projections for New Zealand: climatology and extremes, Weather and Climate Extremes, https://doi.org/10.1016/j.wace.2025.100784.

   Thank you and we agree with your comment. We have now updated the paper with more recent assessments for this section based on Gibson et al. (2025) as follows:

   *"Recent high-resolution (~12 km) downscaled climate projections for New Zealand, based on six global climate models (GCMs) and three regional climate models (RCMs), project a national annual mean warming of 3.1°C (range: 2.0–3.8°C) by 2080–2099 relative to 1986-2005, under the high-emissions SSP3-7.0 scenario (Gibson et al., 2025). Summer maximum temperatures are expected to increase by 3.9°C on average (range: 2.8–4.8°C), particularly affecting inland North Island and high-elevation areas (Gibson et al., 2025). These findings are qualitatively consistent with earlier CMIP5-based projections (Mullan et al., 2018; King et al., 2018)*

   *Precipitation projections show a distinct seasonal and spatial pattern, with winter and spring rainfall increasing by over 20% in parts of the South Island's west coast, while northern and eastern regions of the North Island are likely to experience reduced rainfall, especially in spring and summer (Gibson et al., 2025). Across much of the country, extreme precipitation events are expected to become more intense but less frequent, occurring over shorter durations – except on the South Island's west coast, where both totals and extremes increase, likely due to topographically enhanced dynamical processes (Gibson et al., 2024). Overall, these projected changes reinforce trends identified in earlier CMPI5-based assessments (Gibson et al., 2025)."*

2. Also when discussing changes in climate, the reference period needs to be included; e.g. "expected to warm by 1 degC by 2040 relative to the 1986-2005 average". In addition,

as currently written only a single scenario has been explored (probably King 2010's mid-range scenario), but this choice needs to be highlighted and the scenario information included for context.

Thank you for pointing this out. We have now included the reference period (1986-2005) in the text and added the paper described in point #1 above.

3. In the context of this dataset, it might also be of interest to describe how NZ's climate has changed over the past 60 years (for example referencing the 'seven station series' (https://niwa.co.nz/climate-and-weather/nz-temperature-record/seven-station-series-temperature-data) or the following recent research on changing climate normals, although there are many other studies that could be used.

Srinivasan, R., et al. (2024). Moving to a new normal: Analysis of shifting climate normals in New Zealand. International Journal of Climatology, 44(10), 3240–3263. https://doi.org/10.1002/joc.8521

We appreciate the reviewer's suggestion. We have added a concise summary of New Zealand's observed climate change over recent decades to provide context as follows:

*"New Zealand's climate has warmed by +0.91°C from 1909 to 2009, as shown by the nationally representative Seven Station Series (Mullan et al., 2010). Recent analyses also indicate significant shifts in temperature and precipitation normals across regions and seasons, reflecting both natural variability and anthropogenic climate change (Srinivasan et al., 2024)."*

References:

Mullan, A.B; Stuart, S.J; Hadfield, M.G; Smith, M.J (2010). Report on the Review of NIWA's 'Seven-Station' Temperature Series NIWA Information Series No. 78. 175 p.

Srinivasan, R., Carey-Smith, T., Wang, L., Harper, A., Dean, S., Macara, G., Wang, R. and Stuart, S., 2024. Moving to a new normal: Analysis of shifting climate normals in New Zealand. *International Journal of Climatology*, *44*(10), pp.3240-3263.

4. Figure 4 and related text: Fig 4a units should be in %. Coefficient of variation is a normalised ratio of SD to MEAN, so either expressed as a fraction or %. The incorrect units are also in Table 3.

Thank you for pointing this out. We agree and have corrected the units in Figure 4a and Table 3. The coefficient of variation is now properly expressed as a percentage, reflecting its definition as the ratio of standard deviation to mean.

5. Line 253: Related to above, the sentence "Rainfall variability is highest in the eastern parts of both of the North and South Island" is not correct. Annual rainfall variability **relative to the annual mean** is highest in east (i.e. coefficient of variation). Variability (e.g. variance or standard deviation) of annual rainfall is greatest in the higher elevation parts of the West Coast. Without showing a plot of annual mean rainfall, a reader

unfamiliar with NZ, might come away from this part of the paper not realising that the highest rainfalls are on the west. I recommend that Fig 4 also include a map of mean annual rainfall.

Thank you, we agree that the original wording was misleading. We have revised the sentence to distinguish between absolute variability (e.g. standard deviation) and relative variability (coefficient of variation). A map of mean annual rainfall has now been added to Figure 4 to provide appropriate spatial context.

[Figure]

"Rainfall variability is highest in the eastern parts of both of the North and South Island. Annual PET shows a similar pattern, with higher values to the north and east and very low values to the west of the Southern Alps (Figure 4b)." The revised text now reads as

*"Figure 4 presents the spatial distribution of (a) mean annual rainfall in mm, (b) annual rainfall variability in %, and (c) mean annual potential evapotranspiration (PET) in mm across the CAMELS-NZ catchments. Relative rainfall variability is highest in the eastern regions of both islands, particularly in the South Island, where lower annual rainfall is observed. In contrast, the western regions exhibit lower relative variability but much higher rainfall totals, as indicated in the mean annual rainfall map (Fig. 4a). This reflects New Zealand's distinct west to east hydroclimatic gradient, where prevailing westerlies and orographic effects produce consistently high rainfall in the west and more variable, drier conditions in the east (Tait et al., 2006)"*

---

## Author Comment (AC2)

Thank you very much for providing detailed comments, which allowed us to refine the manuscript and the dataset.

Please see the comments provided by the referee in black font, and our point-by-point response in blue font.

**Anonymous Referee #2**

I agree with the other comments posted about this manuscript— this new open-access dataset will be a great new resource for those interested in New Zealand hydrology and climatology. Similarly, I also find the manuscript to be well written and presented, and have only minor changes to suggest. These are as follows:

We would like to thank Referee #2 for the positive feedback and the insightful comments. Please find our detailed responses to each point below (blue font).

1. Introduction: The recency of its publication is probably the reason for this omission, but the ROBIN (Reference observatory of basins for international hydrological climate change detection) data set (Turner et al. 2025; https://doi.org/10.1038/s41597-025-04907-y) needs to be acknowledged here. As the name indicates, this is a global dataset of streamflow records that have minimal direct anthropogenic influence. Importantly, this includes 111 flow records for New Zealand. Although not diminishing the contribution that this new CAMELS-NZ dataset provides, it is important to note that another large streamflow dataset for NZ exists. Furthermore, it would be very helpful if an additional attribute could be added to this CAMELS dataset to indicate which of its records have passed the ROBIN standards around direct human influence.

   Thank you and we agree that the ROBIN data set is an important global data set that should be mentioned in our manuscript. We have therefore included a short description in Section 2, highlighting also the differences between CAMELS and ROBIN, which mainly lie in the temporal distribution:
   "It goes beyond global data sets such as ROBIN (Turner et al., 2025), which also include catchments in New Zealand (in ROBIN 111 catchments are included), but either have a lower temporal resolution and/or include much fewer catchments than the current CAMELS-NZ."

   As for the application of the classification of ROBIN in either Level 1 or Level 2, it is difficult to apply it here as the criteria are soft. For example, "artificial influences on the catchment minimal or at least stable although this should be based on judgement and local expertise" is hard to verify for the catchments. However, based on a previous study by Cattoën (2025), we have included indicators for each catchment if it is natural, controlled or abstracted. Around 300 catchments are seen as 'natural' and thus would probably belong to Level 1.
   As for the record length, we have around 100 stations that would be Level 1 stations, and around 200 stations that are Level 2 stations.

2. Line 73 and elsewhere: it seems a bit odd for the paper to cite itself – surely this is unnecessary?

This reference was meant to cite the data set itself, stored in the online repository at University of Canterbury, and requested by the journal. Indeed, there was a mistake in the citation pointing to the journal, which has now been corrected. We include this reference (similar to previous CAMELS papers) to provide the access link for the readers.

3. Line 87: Snelder & Biggs (2002) is not the best citation here, at least not in isolation. Although it has some relevance, it does not directly describe the nature of NZ gradients in climate, topography and geology, or the extent to which they are 'remarkable'.

Thank you for pointing this out. We have added two references (Collins, 2020 and McMillan et al., 2016) that both provide details regarding these aspects.

4. Lines 103-104: volcanic catchments might be typical of Pacific Islands, but alpine settings are not.
We agree that this formulation was misleading, as New Zealand indeed has some unique properties among the Pacific islands. We have now stated it more generally: "Secondly, New Zealand's catchments offer valuable insights into hydrological processes typical for Pacific Island nations, e.g. volcanic catchments."

5. Line 114: Naming conventions. While Aotearoa is commonly added to New Zealand (i.e. Aotearoa New Zealand) for the name of the country, it is not the official name of the country. Contrastingly, the North and South Islands do have official names in Te Reo, according to the NZ Geographical Board: Te Ika-a-Māui and Te Waipounamu, respectively (https://gazetteer.linz.govt.nz/). Further, use of Aotearoa New Zealand vs just New Zealand in the manuscript is not consistent. Perhaps use the Te Reo plus English versions could be given at first use (for all NZ place names), then only English thereafter?

Thank you for pointing this out. We have followed this suggestion and added the Te Reo names upon first appearance in the text (once in the abstract, once in the main body of the manuscript), but not thereafter.

6. Line 140: Glacier formation is not quite the right term—this is a process that takes many years (and probably is not occurring anywhere in the world right now?). Perhaps glacier mass balance would be more appropriate.

We appreciate this suggestion for a much better formulation, which was incorporated as suggested.

7. Line 143: state the emissions scenario this projection corresponds to. Same comment for the following precipitation projections. Note that the King (2010) report has also been superseded by more recent assessments.

Thank you and we agree with your comment. We have now updated the paper with more recent assessments for this section based on Gibson et al. (2025) as follows:

*"Recent high-resolution (~12 km) downscaled climate projections for New Zealand, based on six global climate models (GCMs) and three regional climate models (RCMs), project a national annual mean warming of 3.1°C (range: 2.0–3.8°C) by 2080–2099*

*relative to 1986–2005, under the high-emissions SSP3-7.0 scenario (Gibson et al., 2025). Summer maximum temperatures are expected to increase by 3.9°C on average (range: 2.8–4.8°C), particularly affecting inland North Island and high-elevation areas (Gibson et al., 2025). These findings are qualitatively consistent with earlier CMIP5-based projections (Mullan et al., 2018; King et al., 2018)*

*Precipitation projections show a distinct seasonal and spatial pattern, with winter and spring rainfall increasing by over 20% in parts of the South Island's west coast, while northern and eastern regions of the North Island are likely to experience reduced rainfall, especially in spring and summer (Gibson et al., 2025). Across much of the country, extreme precipitation events are expected to become more intense but less frequent, occurring over shorter durations – except on the South Island's west coast, where both totals and extremes increase, likely due to topographically enhanced dynamical processes (Gibson et al., 2024). Overall, these projected changes reinforce trends identified in earlier CMPI5-based assessments (Gibson et al., 2025)."*

References:

Peter Gibson, et al., 2025, Downscaled CMIP6 future climate projections for New Zealand: climatology and extremes, Weather and Climate Extremes, https://doi.org/10.1016/j.wace.2025.100784.

8. Line 249: Given that this paper describes a dataset, more specific information on estimation of PET would be helpful. Conventionally, the Priestley-Taylor equation uses an empirical constant to model the effects of vapour pressure deficit on evaporation, meaning that relative humidity data are not used. Similarly, net radiation (rather than temperature) data are used. However, the Clark et al. (2008) study that is cited here states that "radiation terms are estimated empirically", followed by a citation to Shuttleworth (1993). Correspondingly, it would be helpful to state explicitly how PET is calculated for this data set, and thus how it differs from the conventional Priestley-Taylor method.

We thank the reviewer for their helpful comment. The PET calculation used in this dataset follows a modified Priestley–Taylor method, as implemented in Clark et al. (2008) and McMillan et al. (2016), and differs slightly from the conventional formulation.

Net radiation is estimated as the sum of net shortwave radiation (adjusted for surface albedo) and net longwave radiation, the latter computed empirically using surface air temperature, vapour pressure (from dew point temperature), and a cloudiness factor, following Shuttleworth (1993). Ground heat flux is assumed negligible.

The PET equation uses the Priestley–Taylor form (Equation 14 in Priestley & Taylor, 1972), with alpha = 1.26 for humid conditions. The slope of the saturation vapour pressure curve (Equation 4.2.3 in Shuttleworth, 1993) and the psychrometric constant (Equation 4.2.28) are both explicitly calculated from air temperature, thus making temperature a key input. While relative humidity is not directly used, its influence is implicit via dew point estimates for vapour pressure.

We have updated the manuscript to clarify these points and to distinguish this implementation from the standard Priestley–Taylor method as follows:

*"Potential evapotranspiration (PET) was calculated using a modified Priestley–Taylor method (Priestley & Taylor, 1972), following the implementation in Clark et al. (2008) and McMillan et al. (2016). Net radiation was estimated as the sum of net shortwave radiation (adjusted for albedo) and net longwave radiation, computed empirically using air temperature, vapour pressure (from dew point temperature), and a cloudiness factor (Shuttleworth, 1993). The ground heat flux term was assumed negligible. The slope of the saturation vapour pressure curve and the psychrometric constant were calculated using Equations 4.2.3 and 4.2.28 in Shuttleworth (1993), both of which are explicit functions of air temperature. The Priestley–Taylor coefficient (\alpha) was set to 1.26, appropriate for humid conditions. While relative humidity was not used directly, it affects vapour pressure as dew point is derived from relative humidity. This approach offers a computationally efficient and physically grounded estimate of PET, suited to large-scale hydrological modelling in New Zealand's climatic conditions."*

References:

Priestly and Taylor (1972) "On the assessment of surface heat flux and evaporation using large-scale parameters" Monthly Weather Review, 100, 81-91.

Shuttleworth WJ. Evaporation. In: Maidment DR, editor. Handbook of hydrology. New York: McGraw-Hill; 1993 [chapter 4]. https://hydrology.usu.edu/dtarb/cee6400/ShuttleworthHandbookofHydrologyCh41993.pdf

9. Line 224-225. It would be more accurate to state that whilst a very dense gauging network would be needed to capture the high spatial variability in rainfall for the mountainous regions of NZ, such a network does not currently exist.

   We agree that the previous formulation was misleading, and we changed it as suggested.

10. Line 226-227. While the VCSN does provide these data, access to the VCSN is largely restricted behind a paywall (https://data.niwa.co.nz/products/vcsn-timeseries?_gl=1*1q6gqoh*_ga*MTg1MTUxMzY0OS4xNzUwODE2ODg1*_ga_4CXN46915J*czE3NTA4MTY4ODUkbzEkZzAkdDE3NTA4MTY4OTIkajUzJGwwJGgw ). If this new CAMELS-NZ dataset effectively bypasses that paywall for these catchment-average time series, this is good news and should be more clearly noted!

    Thank you. We would like to point out that the VCSN is gridded data at 5km resolution across the whole country. We are providing bias-corrected catchment averaged information at the selected 369 sites, bias-corrected to the long-term water balance of the catchment-average timeseries.

11. Some comment on the use of catchment average rainfall should be provided, particularly in regions of high rainfall gradients such as the eastern side of the Southern Alps Main Divide. In regions such as this where mean annual rainfall can drop by an order of magnitude across a catchment, how informative is catchment average rainfall? Perhaps an additional data set attribute quantifying this rainfall gradient would be helpful?

We agree that rainfall in New Zealand is quite diverse and cannot be captured by a single characteristic. Besides the rainfall time series, we therefore now provide in the updated version two rainfall characteristics: mean annual rainfall and the coefficient of variation of rainfall for each catchment. This covers not only the magnitude of rainfall but also the gradient in terms of variability. In addition, we provide the number of days per year with high amount of rainfall (usDaysRainGT25, usRainDays10) to also better understand the inter-annual variability.

12. Finally with respect to the VCSN – previous studies have highlighted issues problems with these data with respect to interpolation across complex terrain using a sparse observational network, e.g. Tait and Macara (2014; https://doi.org/10.2307/26169743) and Tait et al. (2012; https://www.jstor.org/stable/43944886). These need to be acknowledged here.

We thank the reviewer for highlighting this important point. We have modified the paper to acknowledge this issue as follows:

*"While the VCSN offers valuable national coverage, previous evaluations have noted interpolation errors in areas of complex terrain due to sparse observational coverage (Tait and Macara, 2012; Tait et al. 2012). Additionally, the number of contributing stations has declined significantly overtime, ranging from a peak of approximately 1,400 stations between 1970 and 1986 to fewer than 400 stations in recent years."*

References:

Tait, Andrew, and Gregor Macara. "Evaluation of Interpolated Daily Temperature Data for High Elevation Areas in New Zealand." *Weather and Climate* 34 (2014): 36–49. https://doi.org/10.2307/26169743.

Tait, Andrew, James Sturman, and Martyn Clark. "An Assessment of the Accuracy of Interpolated Daily Rainfall for New Zealand." *Journal of Hydrology (New Zealand)* 51, no. 1 (2012): 25–44. http://www.jstor.org/stable/43944886.

---

## Author Comment (AC3)

Thank you very much for providing detailed comments, which allowed us to refine the manuscript and the dataset.

Please see the comments provided by the referee in black font, and our point-by-point response in blue font.

**Community Comment #1 (Sacha Ruzzante)**

Overall this seems like a valuable contribution to the growing number of CAMELS datasets. However, I have some suggestions to improve the usefulness of the data and to improve consistency with other CAMELS datasets.

We would like to thank Sacha Ruzzante for providing the comments and suggestions to our manuscript. In this CAMELS data set, we tried to include as many local information on the catchments as possible, while avoiding the inclusion of global data sets, which are already available to everyone. We have pointed this out below, in our detailed responses to each comment (blue font).

1. Can you include time series of glacier evolution, as was done for Camels-CH (Höge et al., 2023)? Or at minimum, have a static attribute that describes glacier cover for each catchment.

   We agree that glacier evolution is important for understanding hydrological processes in glacier-influenced catchments. Although local time series of glacier evolution do exist, they are currently under review and will become available in due course. An updated glacier inventory for New Zealand has been derived from aerial photographs taken between 1978 and 2016 (Baumann, 2021), which provides snapshots of glacier extent over time. Additionally, several studies have used digital elevation models to reconstruct changes in individual glaciers (e.g., the Fox and Franz Josef glaciers; Wang & Kääb, 2015), and more recent work (White, 2024) has drawn on global datasets for glacier thickness, velocity (Millan et al., 2022), and elevation change (Hugonnet et al., 2021). However, while these global datasets capture broad long-term trends, they tend to be less reliable for the small, fast-changing glaciers typical of New Zealand. We recognise the value of including locally derived time series in the future, but for this release, we are not in a position to include glacier evolution time series as was done for CAMELS-CH.

   References:

   Hugonnet, R., McNabb, R., Berthier, E. *et al.* Accelerated global glacier mass loss in the early twenty-first century. *Nature* **592**, 726–731 (2021). https://doi.org/10.1038/s41586-021-03436-z

   Millan, R., Mouginot, J., Rabatel, A. *et al.* Ice velocity and thickness of the world's glaciers. *Nat. Geosci.* **15**, 124–129 (2022). https://doi.org/10.1038/s41561-021-00885-z

   Wang, Di, and Andreas Kääb. 2015. "Modeling Glacier Elevation Change from DEM Time Series" *Remote Sensing* 7, no. 8: 10117-10142. https://doi.org/10.3390/rs70810117

   White, Rebecca Margaret. "An investigation into the effects of Proglacial Lakes on

Mountain Glacier Dynamics in New Zealand." PhD diss., University of Leeds, 2024. oai:etheses.whiterose.ac.uk:36693

2. Some of the static attributes are provided as categorical variables that indicate the dominant category (eg. land cover, geology). For many applications it is more useful to know the percentage of the catchment that falls into each category, rather than just the most common category.

We agree that the knowledge on the precise percentage could be useful for many applications. Unfortunately, this information is not available from the two sources we have considered (FENZ and REC data sets). A different geological classification for New Zealand which could be used to derived percentage contribution would be available from GNS (https://data.gns.cri.nz/geology/). Since this classification deviates from the current one and also provides much more classes, we decided to keep the simpler one to avoid overly complicated attribute tables. However, we have added the reference to the manuscript for interested readers.

3. There are many useful static attributes that can be calculated but are not currently included. For example, soil characteristics from SoilGrids (Poggio et al., 2021), catchment average elevation, mean annual temperature, precipitation seasonality, etc. See other camels datasets or the Caravan project (Kratzert et al., 2023) for examples.

Thank you for your suggestions. In developing this CAMELS dataset, we aimed to prioritise regionally relevant and locally sourced information, rather than replicating global datasets such as SoilGrids, which are already widely accessible and may not perform well in highly heterogeneous terrain where terrain complexity and low station density limit model accuracy like Aotearoa New Zealand. For example, global soil datasets are known to have limited accuracy in steep, mountainous terrain where soil observations are sparse.

Catchment average elevation is indeed included (labelled as "elevation"), and we appreciate you pointing out the earlier mislabelling as "elevation of the gauge" – this has now been corrected in the table.

With respect to mean annual temperature and precipitation seasonality: while these attributes may be used in other CAMELS datasets, we opted not to include them in this release for several reasons. First, mean annual temperature can be readily derived by users from the provided hourly or daily time series, and its meaning in a highly orographically complex region like New Zealand is less straightforward than in flatter continental regions. Second, precipitation in New Zealand is generally distributed relatively evenly throughout the year, with only moderate seasonality in most areas. As such, calculating a meaningful and comparable seasonality index would require first defining wet and dry seasons across diverse hydroclimates, which may not be robust or useful at a national scale.

We recognise the value of including harmonised attributes for cross-regional model benchmarking (e.g., via Caravan), and we are open to extending the attribute set in future versions – ideally with New Zealand-specific data sources and derived attributes that reflect the unique hydrometeorology of the region.

4. Are there other climate datasets that could be included as well? For machine learning models previous work has shown that including several climate datasets in trainingF usually improves overall model skill. For example, ERA5-Land (Muñoz Sabater, 2019), the New Zealand Reanalysis Dataset (Pirooz et al., 2023) CHIRPS (Funk et al., 2015), CPC (Chen & Xie, 2008), etc. You may want to look at how these were included in other camels datasets such as Camels-BR (Chagas et al., 2020). Some of these provide daily data only, but that is what many users will want anyway. For snow-affected catchments it would be useful to have a SWE product (eg. ERA5-Land).

Thank you very much for this suggestion to expand the local data with global datasets. Expanding the local dataset by including additional data from e.g. ERA5-Land or CHIRPS might be challenging, as it can be seen as an endorsement of such data in terms of their quality and applicability in the local context. For example, Queen et al. (2023) used ERA5 data in their work in the New Zealand context with less convincing results. We feel that data users that are keen to add such data to their analyses should obtain such publicly available datasets on their own and research if the level of accuracy suits their needs.

Reference: Queen, L. E., S. Dean, D. Stone, R. Henderson, and J. Renwick, 2023: Spatiotemporal Trends in Near-Natural New Zealand River Flow. J. Hydrometeor., 24, 241–255, https://doi.org/10.1175/JHM-D-22-0037.1.

5. It would be useful to also provide daily aggregated streamflow and meteorology data. Most hydrologic models are built on daily data, and for benchmarking models across different research groups it is useful to know that everyone is using exactly the same data. Providing the daily aggregated data helps ensure this.

Thank you for the suggestion. We agree that providing daily aggregated streamflow and meteorological data will improve consistency and usability, especially for benchmarking across different modelling frameworks. We have included the daily aggregated datasets to support and facilitate broader adoption of the data.

6. The paper states "Information on how to obtain permission [for the 13 gauges that require it] is provided in the readme file", but this is missing from the readme file.

The readme-file is available in the streamflow folder. We have updated the information on its location in the manuscript to ensure the availability of this information.

7. I'm not sure what the authors mean by the "original temporal structure" in the following:"All time series data are reported in the local time zone, and include the effects of daylight saving time (DST) where applicable. No corrections or transformations were applied to standardise timestamps across the dataset. This decision was made to preserve the original temporal structure of the observations." It would be more useful if all timestamps were provided in standard time, and it is quite possible to do this while preserving the temporal structure of the data.

Thank you, we have updated the time series to use New Zealand Standard Time (NZST which is UTC+12 hours).

8. There are some negative streamflow values. For example, station 29231, which has a number of timestamps with flow of -0.003 cms. What does this mean?

Thank you for pointing this out. We have reviewed the data and found that, in addition to the designated missing value of -9999 (which was correctly handled), a small number of other negative values such as -0.003 m³/s were mistakenly retained during the conversion from NetCDF to CSV. Since streamflow cannot be negative, these values have no physical meaning. We have now updated the dataset to remove all such values and replaced them with NA to correctly indicate missing or invalid data.

**References:**

Chagas, V. B. P., Chaffe, P. L. B., Addor, N., Fan, F. M., Fleischmann, A. S., Paiva, R. C. D., & Siqueira, V. A. (2020). CAMELS-BR: Hydrometeorological time series and landscape attributes for 897 catchments in Brazil. Earth System Science Data 12(3), 2075–2096. https://doi.org/10.5194/essd-12-2075-2020

Chen, M., & Xie, P. (2008, January 8). CPC unified gauge-based analysis of global daily precipitation. Western Pacific Geophysics Meeting, Cairns, Australia.

Collins, D. B. G.: New Zealand river hydrology under late 21st century climate change. Water, 12(8), 2175. https://doi.org/10.3390/w12082175, 2020.

McMillan, H., Booker, D., and Cattoën, C.: Validation of a national hydrological model, Journal of Hydrology, 541, 800–815, https://doi.org/10.1016/j.jhydrol.2016.07.043, 2016.

Funk, C., Peterson, P., Landsfeld, M., Pedreros, D., Verdin, J., Shukla, S., Husak, G., Rowland, J., Harrison, L., Hoell, A., & Michaelsen, J. (2015). The climate hazards infrared precipitation with stations—A new environmental record for monitoring extremes. Scientific Data, 2(1), 150066. https://doi.org/10.1038/sdata.2015.66

Höge, M., Kauzlaric, M., Siber, R., Schönenberger, U., Horton, P., Schwanbeck, J., Floriancic, M. G., Viviroli, D., Wilhelm, S., Sikorska-Senoner, A. E., Addor, N., Brunner, M., Pool, S., Zappa, M., & Fenicia, F. (2023). CAMELS-CH: Hydro-meteorological time series and landscape attributes for 331 catchments in hydrologic Switzerland. Earth System Science Data, 15(12), 5755–5784. https://doi.org/10.5194/essd-15-5755-2023

Kratzert, F., Nearing, G., Addor, N., Erickson, T., Gauch, M., Gilon, O., Gudmundsson, L., Hassidim, A., Klotz, D., Nevo, S., Shalev, G., & Matias, Y. (2023). Caravan—A global community dataset for large-sample hydrology. Scientific Data, 10(1), 61. https://doi.org/10.1038/s41597-023-01975-w

Muñoz Sabater, J. (2019). ERA5-Land monthly averaged data from 1950 to present [Dataset]. Copernicus Climate Change Service (C3S) Climate Data Store (CDS). https://doi.org/10.24381/cds.68d2bb30

Pirooz, A., Moore, S., Carey-Smith, T., Turner, R., & Su, C.-H. (2023). The New Zealand Reanalysis (NZRA): Development and preliminary evaluation. Weather and Climate, 42(1), 58–74. https://doi.org/10.2307/27226715

Poggio, L., de Sousa, L. M., Batjes, N. H., Heuvelink, G. B. M., Kempen, B., Ribeiro, E., & Rossiter, D. (2021). SoilGrids 2.0: Producing soil information for the globe with quantified spatial uncertainty. SOIL, 7(1), 217–240. https://doi.org/10.5194/soil-7-217-2021

---

## Author Comment (AC4)

Thank you very much for providing detailed comments, which allowed us to refine the manuscript and the dataset.

Please see the comments provided by the referee in black font, and our point-by-point response in blue font.

**Community Comment #2 (Ather Abbas)**

I am the author of AquaFetch (https://github.com/hyex-research/AquaFetch), a Python library designed to unify open-source hydrological datasets within the Python environment. While integrating CAMELS-NZ into our package, we encountered a few issues and would greatly appreciate it if they could be addressed in a future release of the dataset:

Thank you for your thoughtful feedback and for incorporating CAMELS-NZ into the AquaFetch library. It's great to see the dataset contributing to open hydrological research. Please find our replies to your comments below (in blue font):

- The streamflow file for station 74368 is empty.

  Thank you for pointing this out. The streamflow file for station 74368 is intentionally empty because access to this data requires permission. This detail, however, was missing from the README file. We have now updated the README to clearly indicate the restricted access for this station's streamflow data.

- The index/datetime format in the streamflow file for station 57521 differs from that of the other stations.

  Although the datetime format appeared consistent initially, we discovered a formatting difference when inspecting the file. This has been fixed now.

- The time series files for all stations and all five features contain missing and duplicated datetime indices. This appears to be due to data missing or duplicated at the 02:00:00 timestamp on certain dates.

  Thank you for your comment. This is because timestamps were in local time using New Zealand Daylight Savings. We have now updated all time series to use New Zealand Standard Time (NZST) which is UTC + 12 hours to remove this issue of missing and duplicated datetime indices.

- Adding a new attribute to the shapefile to identify the Station_ID for each catchment polygon would be highly beneficial.

  Thanks for the suggestion.  We have taken this into consideration and have added Station_ID attribute for each catchment polygon.

Thank you very much for making this dataset available to the open-source community.